# Morphology and structure of ZIF-8 during crystallisation measured by dynamic angle-resolved second harmonic scattering

Stijn Van Cleuvenbergen[1], Zachary J. Smith [2], Olivier Deschaume [3], Carmen Bartic[3], Sebastian Wachsmann-Hogiu[4,5], Thierry Verbiest[1] & Monique A. van der Veen [6]

Recent developments in nonlinear optical light scattering techniques have opened a window into morphological and structural characteristics for a variety of supramolecular systems. However, for the study of dynamic processes, the current way of measuring is often too slow. Here we present an alternative measurement scheme suitable for following dynamic processes. Fast acquisition times are achieved through Fourier imaging, allowing simultaneous detection at multiple scattering angles for different polarization combinations. This allows us to follow the crystal growth of the metal organic framework ZIF-8 in solution. The angle dependence of the signal provides insight into the growth mechanism by probing the evolution of size, shape and concentration, while polarization analysis yields structural information in terms of point group symmetry. Our findings highlight the potential of dynamic angle-resolved harmonic light scattering to probe crystal growth processes, assembly–disassembly of biological systems, adsorption, transport through membranes and myriad other applications.

[1] Department of Chemistry, Molecular Imaging and Photonics, KU Leuven, Celestijnenlaan 200D, 3001 Heverlee, Belgium. [2] Department of Precision Machinery and Precision Instrumentation, University of Science and Technology of China, 443 Huangshan Road, 230027 Hefei, Anhui, China. [3] Department of Physics and Astronomy, Soft-Matter Physics and Biophysics Section, KU Leuven, Celestijnenlaan 200 D - box 2416, B-3001 Heverlee, Belgium. [4] Department of Bioengineering, McGill University, Montreal H3A 0E9, Canada. [5] Department of Pathology and Laboratory Medicine and Center for Biophotonics, University of California Davis, Sacramento CA 95817 CA, USA. [6] Catalysis Engineering, Department of Chemical Engineering, Delft University of Technology, 2629 Delft, The Netherlands. Correspondence and requests for materials should be addressed to S.V.C. (email: stijn.vancleuvenbergen@kuleuven.be) or to M.A.v.d. V. (email: m.a.vanderveen@tudelft.nl)

Gaining control over crystallization processes is a long-standing ambition of crystal growers. Hitherto process optimization often relies exclusively on characterization of the final product, assuming that this gives an unambiguous connection to solution processes such as nucleation and growth[1]. It has however become increasingly clear that crystal growth in solution is much more complex than that, often involving processes such as nonclassical nucleation, structural transformations through transient intermediates, or the occurrence of metastable phases[1–3]. In order to gain control over crystallization in a truly rational way an intimate knowledge of the crystallization process in terms of size, shape, concentration, and structure is essential during all stages of crystal formation, requiring time-resolved in situ studies during growth.

Second-order nonlinear optical techniques might be particularly interesting in this respect since they are inherently sensitive to the way matter is organized. The second-order nonlinear optical response of a material strongly depends upon its characteristic symmetry elements[4,5]. This is illustrated most drastically by the effect of an inversion center, leading to an effective cancellation of the response. Other symmetry elements will determine how a material responds to polarized light. For non-centrosymmetric structures (i.e., lacking inversion symmetry), second-order measurement techniques thus allow analysis in terms of point group symmetry and orientation in a noninvasive manner. Second harmonic generation is arguably the most well-known second-order process, combining two photons at frequency $\omega$ to a single new photon at the double frequency $2\omega$ by interaction with a nonlinear material. Lately, second harmonic generation techniques have become increasingly popular for the characterization of crystalline structures of organics[6–8], pharmaceuticals[9–11], proteins[12–14], and MOFs[15–18].

Similarly second harmonic scattering (SHS) can be employed to interrogate the structure of species in solution. For scatterers that are small compared to the wavelength (<10 nm for wavelengths typically used), like molecules, aggregates, or nanocrystals, SHS is incoherent and generally termed hyper-Rayleigh scattering[19]. Polarization-resolved HRS studies allow measurement of symmetry properties of molecules and aggregates[4,20], and can be used to study structural correlations in liquids[21,22]. Time-resolved hyper-Rayleigh scattering has even been used to monitor the crystal growth of ZnO and iron iodate nanocrystals with millisecond time resolution, giving access to nucleation, growth, and ripening rates[23,24]. These studies however did not include polarization analysis so no insight into structural evolution during growth could be inferred, and since HRS shows no (in-plane) angle dependence, measurements were performed at a single angle only. Beyond the Rayleigh limit (i.e., for species larger than ~10 nm), phase relationships between different points in the structure give rise to coherent effects, resulting in an angle-dependent signal. Like for static light scattering, angle-resolved second harmonic scattering (AR-SHS) can measure the size, shape and concentration of scattering species by measuring the signal intensity at different scattering angles[25,26]. Through polarization-resolved measurements, AR-SHS additionally probes the symmetry of the scattering species[27]. AR-SHS, and its nondegenerate sum frequency scattering analogue AR-SFS, have hitherto mainly been used for characterizing the surface of otherwise centrosymmetric structures, where inversion symmetry is broken by default. Polystyrene particles with malachite green dye molecules adsorbed on their surface have served as a favorite model system in numerous nonlinear light scattering studies, giving access to particle size, surface concentration, and molecular conformation[25,28,29]. Likewise AR-SFS measurements have

emerged as a powerful tool to investigate the interfacial structure of nanoscopic micelle and droplet interfaces[30–33]. On two occasions AR-SHS has been employed to measure bulk non-centrosymmetric crystalline domains in a further amorphous polymer matrix[34,35]. The angle-resolved scattering patterns could be fitted to obtain the size of the buried domains, while polarization-resolved measurements were able to probe the crystal structure.

As AR-SHS can simultaneously measure size, concentration, and symmetry in solution, it is ideally suited to follow crystal growth in situ. Yet, the current way of measuring AR-SHS patterns by rotating a goniometric arm in discrete steps around the cuvette and measuring angle by angle is too time consuming to follow dynamic processes[25,36]. In this work we develop an alternative way of measuring AR-SHS patterns (i.e., dynamic AR-SHS or dAR-SHS) by imaging a large portion of the AR-SHS pattern on an electron multiplying charge coupled device (EM-CCD) camera in a Fourier-imaging scheme, allowing for single-shot measurements of the second harmonic signal at multiple angles which significantly boosts time resolution. In order to benchmark the technique we first measured a well-known system, a series of polystyrene beads of different sizes with adsorbed malachite green molecules, for which the obtained angular scattering patterns were in good agreement with the theoretically expected curves. dAR-SHS was then employed to follow the crystallization of the well-known MOF ZIF-8 in situ. The obtained dAR-SHS curves were fitted to an appropriate model which allowed extracting size, shape, and concentration information during growth. By adding a polarizer in the detection path, different polarization combinations could be measured simultaneously, giving access to symmetry information during crystal formation. The polarization signature of the detected particles was in agreement with the $T_d$ point group symmetry of phase-pure ZIF-8. Simultaneous analysis towards size and concentration moreover revealed a change in growth mechanism, in line with an earlier study employing static light scattering, from particle coalescence to growth by monomer addition[37]. Note that this is the first nonlinear optical scattering study that employs AR-SHS to monitor a dynamic process. Moreover, it additionally makes use of polarized light analysis to monitor structural evolution during crystal formation.

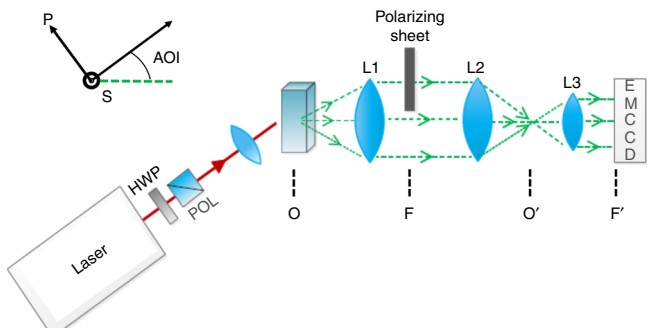

**Fig. 1** Schematical depiction of the experimental setup. O and O′ are object and image planes, respectively, F and F′ are the Fourier plane and its image. The propagation direction and the S and P polarization directions are indicated as well. The fundamental beam is indicated in red, the second harmonic in green. Note that the incident light is aligned under an angle of incidence (AOI) with respect to the optical axis of the collection system. HWP half-wave plate, POL polarizer

## Results

**Fast Fourier-imaging of nonlinear scattering patterns.** The setup is shown schematically in Fig. 1. Complete details of the setup can be found in the Methods section. The angle of incidence is larger than 30° such that, as AR-SHS patterns are symmetric around 0°, we do not measure the same portion of the scattering pattern twice around 0°. In addition, artefacts due to the incident laser light directly entering the collection system are avoided. The detection is realized by simultaneously imaging a large portion of the scattering patterns on a CCD camera in a Fourier-imaging scheme[38–40], resulting in fast acquisition times. The scattered light is collected by a fast collection lens (L1). The angle at which the scattered light enters this lens determines its position in the back focal plane or Fourier plane of this lens (F), and thus an image of the scattering pattern is formed. Subsequently this Fourier plane is imaged by a telescopic system onto an EM-CCD camera. In Fig. 2a, the resultant image of the second harmonic light in the back focal plane scattered by Polystyrene beads coated with malachite green is shown. By using the image of a grating of known pitch a pixel-to angle conversion is achieved (for the full procedure see Methods). The data gathered by the EM-CCD camera are then corrected in three ways (see Supplementary Note 1): first pixel-to-angle conversion is performed, then dark correction, and finally the data are corrected for the difference in collection efficiency at different angles.

To validate the performance of the AR-SHS setup, we measured a series of polystyrene beads of known sizes coated with malachite green dye molecules. This system has served in a wide range of AR-SHS studies[25], since malachite green provides a relatively strong second harmonic response. We measured polystyrene beads of 170, 320, and 490 nm in size, and found that the AR-SHS patterns agree well with the theoretically expected curves according to a Rayleigh-Gans-Debye model developed by Yang et al. (see Supplementary Note 2)[41]. Of particular interest for structural analysis is the polarization dependence of the AR-SHS signal. We adopt the widely used S and P designations to describe the polarization of the fundamental and second harmonic light in the macroscopic coordinate system, as indicated in Fig. 1. The second harmonic intensity $I_{IJK}$ can be evaluated for different polarization combinations. Indices $J$ and $K$ refer to the two fundamental light fields that combine to create the second harmonic light field. In our experiment these

fields are degenerate and S polarized. The first index $I$ corresponds to the polarization of the scattered second harmonic light and can be selected, e.g., S or P, by inserting a polarizer in the detection path, arriving at $I_{SSS}$ or $I_{PSS}$, respectively. By inserting a polarizing sheet in the S direction halfway into the back focal plane, the $I_{SSS}$ polarization combination is selected in the bottom half of the image, as indicated in green in Fig. 2b. Since there is no polarizer present in the detection path for the upper top half of the image, the pattern extracted from the blue area in Fig. 2b corresponds to the sum of the $I_{SSS}$ and $I_{PSS}$ polarization combinations. This way, both $I_{SSS}$ and $I_{PSS}$ can be evaluated simultaneously (for more details see Supplementary Note 2). In Fig. 2c the resulting patterns are shown for 320 nm polystyrene beads coated with malachite green. Note that depending on the system under study, other polarization combinations can provide additional information, for instance on the chirality of the structure[42].

**dAR-SHS during crystal growth of ZIF-8.** The main advantage of measuring dAR-SHS by imaging scattering patterns, is the inherent speed of acquisition. A large portion of the pattern (up to 40° in solution for the optics used in this work) is recorded simultaneously without the need of mechanical motion of detectors or optical parts, resulting in unprecedented time resolution for AR-SHS. In this way, the potential of AR-SHS to measure size, concentration, and symmetry in solution for dynamic processes is unlocked. To demonstrate this, we measured the crystal growth of MOF ZIF-8 in methanolic solution at room temperature.

Metal organic frameworks (MOFs) can be considered as 3D-extended coordination complexes, where multi-dentate organic linker molecules connect metal nodes in a crystalline arrangement. ZIF-8 is a well-known MOF which is prepared by combining a zinc source with 2-methylimidazole linker molecules. Phase-pure ZIF-8 has noncentrosymmetric cubic I-43m space group ($T_d$ point group) symmetry and has a significant effective nonlinear optical coefficient ($d_{eff}$ up to ~0.3 pm/V)[15]. Also note that ZIF-8 is optically transparent in the visible window (absorption around 220 nm)[43]. The synthesis procedure we selected also includes addition of sodium formate, which competitively binds to the zinc centers. This so-called modulated synthesis results in fewer nucleation centers and leads to

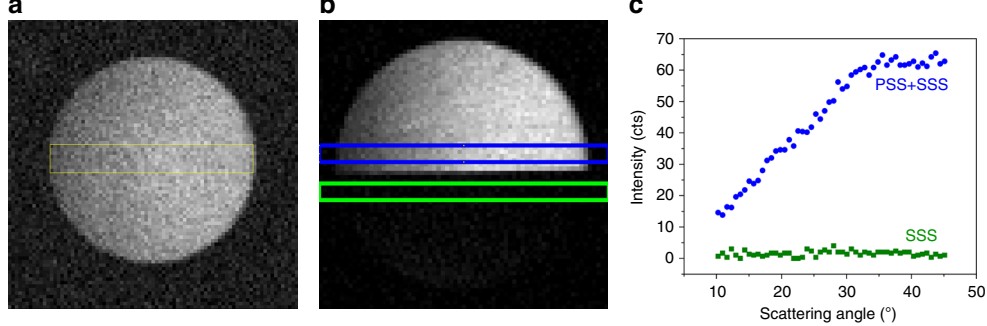

**Fig. 2** Angle-resolved second harmonic scattering patterns obtained through Fourier imaging. **a** Image of the angle-resolved second harmonic scattering (AR-SHS) scattering pattern for 320 nm polystyrene beads coated with malachite green, **b** Image of the AR-SHS scattering pattern for 320 nm polystyrene beads coated with malachite green with insertion of a polarizing sheet along the S-polarization direction) in the lower half of the back focal plane. By symmetry considerations the SSS signal (i.e., both the incident and detected light along the S-polarization direction) originating from the surface of polystyrene beads coated with malachite green vanishes[60], which is why the bottom half of the image appears dark. The blue and green areas depict the regions of interest analyzed in Fig. 2c. **c** Graphs of selected areas render AR-SHS for different polarization combinations simultaneously: blue no polarizing sheet, PSS (i.e., incident light along the S-polarization direction and detected light along the P-polarization direction) + SSS; green polarizing sheet: SSS

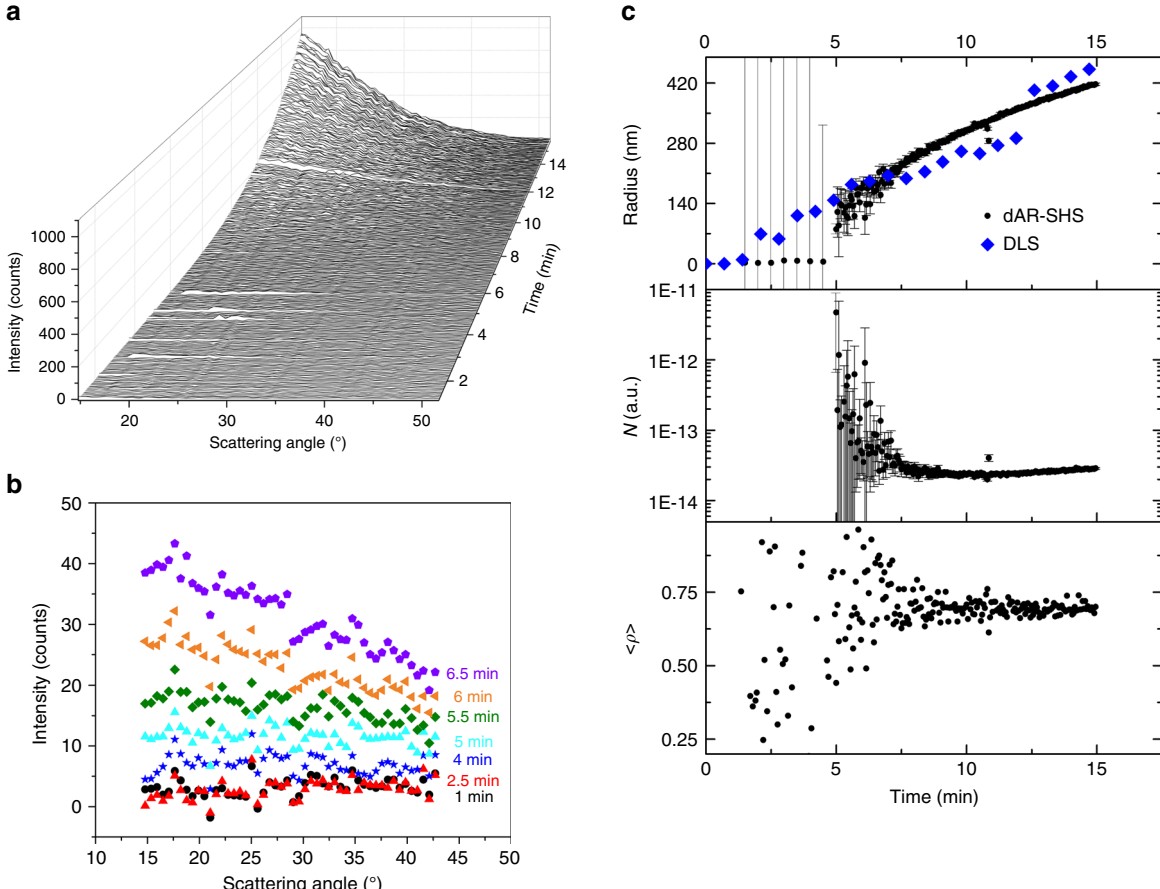

**Fig. 3** Dynamic angle-resolved second harmonic scattering during crystal growth of ZIF-8. **a** Dynamic angle-resolved second harmonic scattering (dAR-SHS) patterns as function of time. During the course of the measurement AR-SHS patterns become more intense and increasingly forward directed as expected for growing particles. **b** AR-SHS patterns at different times during the earliest stages. The curves shown are an average over 10 measurements (~30 s), **c** Radius, number of particles (N) and depolarization extracted from the data presented in Fig. 3a by nonlinear regression towards the nonlinear Rayleigh-Gans-Debye model for monodisperse spheres. The standard errors for the derived parameters are estimated by the OriginPro fitting software according to the Error Propagation formula. In the first panel data obtained from a dynamic light scattering experiment (DLS) are shown in blue in Fig. 3c. The depolarization approaches 0.64, as expected for the symmetry group of ZIF-8 ($T_d$)

micrometer-sized crystals being formed, instead of nanosized (<100 nm) crystals for the unmodulated synthesis[37]. This synthesis will serve as a benchmark here.

The dynamic AR-SHS patterns for the crystal growth of ZIF-8 are shown in Fig. 3a. In line with an in situ static light scattering study by Cravillon et al. we measured the first 15 min, before the onset of sedimentation[37]. The time resolution is 3 s, but since signal levels were very low at the beginning of the measurement an average over 10 measurements was taken for the first 4.5 min. Over time, the second harmonic signal starts building up slowly and as the intensity increases the patterns become more and more forward directed as expected for growing particles. In contrast with surface AR-SHS patterns (as measured for the polystyrene beads coated with malachite green), AR-SHS originating from the bulk is always forward directed and becomes sharper (symmetric around 0°) and more intense as particles grow larger. In Fig. 3b AR-SHS patterns are shown for the earliest stages of the growth process. During the first minutes the signal levels are close to zero (about 1–2 counts). After about 4 min the second harmonic signal increases and in the following minutes the patterns become increasingly forward directed.

In order to derive information from the recorded patterns we adopted a model from de Beer and co-authors, describing the

total scattered intensity as[34]:

$$I_{u_0,u_1,u_2}(r_0) \propto N\varepsilon_{u_1}^2\varepsilon_{u_2}^2 |F(qR)|^2 \int_0^{2\pi}\int_0^{\pi}\int_0^{2\pi} \left|\sum_{xyz} G(\theta;\psi,\xi,\zeta)_{xyz}\right|^2 d\zeta \sin\xi d\xi d\psi \quad (1)$$

With $N$ the number of particles, $\varepsilon$ the electric field amplitude, $\mathbf{e}_{u_i}$ the unit vector for polarization state $u_i$ of incoming ($i = 1, 2$) and outgoing ($i = 0$) light. $I_{u_0,u_1,u_2}(r_0)$ then represents the total scattered intensity for a given polarization state at the detector positioned at $r_0$. The second harmonic intensity depends linearly on the number of particles $N$, giving a link to the concentration in solution. For a degenerate process, $u_1$ equals $u_2$ and hence the second harmonic intensity depends on the fundamental intensity squared ($\sim\varepsilon^4$). $F(qR)$ is the scattering form factor. It relates the intensity to the scattering angle $\theta$ through the scattering wave vector $\mathbf{q}$. It is defined as $\mathbf{q} = (\mathbf{k}_0 - 2\mathbf{k}_\omega)$, with $\mathbf{k}_0$ the wave vector of the scattered second harmonic light and $\mathbf{k}_\omega$ the wave vector of the incident light (see Supplementary Figure 3). The magnitude of the scattering wave vector is then found as $q = 2k_0\sin(\theta/2) = (8\pi n/\lambda)\sin(\theta/2)$, with $k_0$ the magnitude of the wave vector of the scattered second harmonic light, $n$ the refractive index and $\lambda$ the wavelength. $F(qR)$ depends on the shape and size distribution of the scattering particles. $G(\theta;\psi,\xi,\zeta)$ relates the nonlinear optical

properties of the individual crystallites to the macroscopic response. It depends on the scattering angle $\theta$, the nonlinear tensor $\chi^{(2)}$ and the orientation of individual domains defined by Euler angles $\psi$, $\xi$ and $\zeta$. While this model uses the Rayleigh-Gans-Debye approximation, which is strictly speaking not valid here since the refractive index of the solvent ($n \approx 1.33$) differs substantially from that of ZIF-8 ($n \approx 1.6$), it serves as a good approximation as long as the measured particles are relatively small compared to the wavelength. In the following paragraphs, we will discuss in more detail how symmetry, size, shape, and concentration information can be derived from nonlinear scattering experiments based on the results obtained for ZIF-8.

Structural information is embedded in the nonlinear tensor $\chi^{(2)}$ through $G(\theta; \psi, \xi, \zeta)$, which projects the tensor components of the crystalline domain from the local crystal framework to the macroscopic framework:

$$G(\theta; \psi, \xi, \zeta) = \sum_{\alpha_0, \alpha_1, \alpha_2} \chi^{(2)}_{\alpha_0, \alpha_1, \alpha_2} \prod_{i=0}^{2} \left( \Re\left( \boldsymbol{e}'_{\alpha_j} \right) . \boldsymbol{e}_{u_i} \right) \qquad (2)$$

$\alpha_i$ $(x, y, z)$ represents the orthonormal base for crystalline axes $x$, $y$, and $z$. $\Re$ represents a rotation over Euler angles $\psi$, $\xi$, and $\zeta$ describing the crystalline domain for an arbitrary orientation. To arrive at the total scattering intensity in solution, isotropic averaging over all Euler angles $(\psi, \xi, \zeta)$ is performed for all $(N)$ contributing crystallites. Since the number of non-zero tensor components depends on the point group symmetry[4,5], $G(\theta; \psi, \xi, \zeta)$ depends on the symmetry of the probed structure. The same principle is routinely employed in hyper-Rayleigh scattering measurements. Indeed, by setting the scattering angle to 90° and inserting the different components of the molecular hyperpolarizability $\beta$ (instead of the bulk nonlinear susceptibility $\chi^{(2)}$) for polarization combinations $I_{SSS}$ and $I_{PSS}$, one arrives at the well-known expressions for the depolarization $\rho = I_{PSS}/I_{SSS}$. Unlike for linear Rayleigh scattering, which has no off-axis polarization components ($\rho = 0$)[44], the second-order response is always depolarized to some extent. The degree of depolarization is described by the expressions derived by Cyvin et al. in terms of the contributing tensor elements[45]. At a scattering angle of 90°, pure octupoles (e.g., $T_d$, $D_{3h}$, $D_{2d}$,..) and pure dipoles ($C_{\infty v}$ with dominant $\beta_{zzz}$) have a depolarization 2/3 and 1/5, respectively, while for point groups with multiple non-identical tensor elements (e.g., $C_{2v}$), the relative magnitude of different elements must be taken into account[4,45]. Pure phase ZIF-8 has noncentrosymmetric I-43m space group symmetry corresponding to octupolar $T_d$ symmetry. From symmetry arguments it is found that this point group has only one unique bulk nonlinear susceptibility tensor element $\chi^{(2)}_{xyz}$. Since AR-SHS detects scattering angles different from 90°, we derived the expected depolarization depending on the in-plane scattering angle for the $T_d$ point group (I-43m space group) of ZIF-8 in detail in Supplementary Note 3. By averaging the expected value over all scattering angles detected in our experiment, we arrive at an average depolarization $\langle\rho\rangle$ of 0.64 for ZIF-8. The first 5 minutes the measured signal is too low to extract meaningful results. Take into account that the detected SSS signal is substantially lower than the PSS + SSS signal shown in Fig. 3b. Once the signal increases the depolarization converges towards the expected value, which implies that the known ZIF-8 structure is formed.

For analysis of the evolution of particle size the AR-SHS signal was analyzed as a function of time. For the SSS polarization combination particle size can be derived directly from the form factor $F(qR)$ like in linear scattering, although here the SHS wave vector is used. For the analysis, a collection of monodisperse

spheres was assumed (see further) and AR-SHS curves (PSS + SSS) were analyzed towards obtaining the particle radius $R$ by nonlinear regression. The reported standard errors for the derived parameters are estimated according to the Error Propagation formula by the OriginPro fitting software. Note that for AR-SHS measurements of polarization combinations other than SSS, an angle dependence resulting from $G(\theta; \psi, \xi, \zeta)$ must be taken into account as well, as explained in more detail in Supplementary Note 3. Before 5 min the data were averaged over 10 measurements (30 s, outliers removed) and smoothed by adjacent averaging (5 point) prior to fitting, but nonetheless large errors on the fitted values result. The first statistically significant results are extracted after 5 min and the detected particles have a radius of about 100 nm. Over the course of the measurement the particles grow to a radius of about 400 nm. Typical fits of the AR-SHS patterns are shown in Supplementary Figure 5. We repeated the experiment starting from the same stock solutions while monitoring the evolution of size with time-resolved dynamic light scattering (DLS, shown in blue in Fig. 3c), and found that the measured radius is in good agreement for both experiments. DLS detects the first particles after about 2 min compared to 5 min for AR-SHS. It has to be noted here that the size dependence is the same for both techniques ($\sim R^6$). To evaluate the sensitivity of the AR-SHS technique in the earliest stages we performed a standard SHS experiment in transmission mode, as shown in Supplementary Figure 6. In this type of experiment the generated SHS light is collected with a high numerical aperture lens, spectrally resolved by a spectrometer, and then focused on a detector (see Methods). Since light is averaged over a wide range of angles instead of separated into different angles as for AR-SHS, the sensitivity is higher at the expense of losing angle-resolved information. Thus, size and shape cannot be extracted in this manner. The standard SHS experiment can detect meaningful signals from the start of the measurement, since the solution generates a signal even before crystallization is initiated. After about 2 min, the SHS signal rapidly increases, indicating that the first particles are being formed. From this point on the depolarization, which was simultaneously measured, almost instantly shifts from a low value for the solution of 2-methylimidazole in methanol towards the expected value of ~2/3 characteristic of ZIF-8 ($T_d$). This implies that the first particles detected by DLS already adopted the ZIF-8 structure.

Information about crystal morphology and dispersity can be obtained from the AR-SHS curves as well, by fitting to appropriate models. To illustrate the shape selectivity of the method we compared fits of the AR-SHS patterns obtained for the largest particles (15 min) to models assuming a spherical and cylindrical shape. The data were plotted as a function of $q$ as shown in Supplementary Figure 7. For the SSS polarization combination the angle dependence results exclusively from the form factor $F(qR)$, such that well-known functions used in linear light scattering experiments can be applied accordingly (See Supplementary Note 5). The spherical shape model provides a good fit to the data, in line with results obtained by static light scattering[37]. For the cylindrical model the data fit best to a cylinder with aspect ratio around 1, indicating that the morphology of the crystallites is highly symmetrical. The form factor also enables determination of particle size distributions. In Supplementary Figure 6b we fitted the AR-SHS curves at 15 min to a model assuming a Gaussian size distribution for spherical particles. The fitted curves indicate a relatively narrow size distribution (dispersity = $0.22 \pm 0.04$). It is important to remark that both the particle shape and the width of the size distribution mostly affect scattering at higher angles, where fringes start to emerge. In the forward direction and at low angles scattering patterns are roughly the same for different form factors,

corresponding to the pattern expected for surface-area equivalent spheres[46]. The smaller the particles, the more the characteristic fringes shift towards higher scattering angles ($q$-values). This is illustrated by the AR-SHS curves at 12 min shown in Supplementary Figure 6a, which fit well to scattering from a collection of almost monodisperse spheres. While for this experiment we opted for measuring at low angles where the intensity is significantly higher, the fidelity of the analysis of shape and size distribution can therefore be improved by measurements at higher angles.

The scattering intensity depends linearly on the number of particles $N$ (Eq. 1). We fitted the AR-SHS curves to a scaling factor multiplied with the modulus of the form factor squared and the isotropic average over the modulus of $G(\theta; \psi, \xi, \zeta)$ squared. Since the intensity of the fundamental light beam remains constant during the experiment, the scaling factor is linear with respect to the number of particles in the focal volume. If AR-SHS results from a collection of particles of identical (crystal) structure the extracted scaling factor is a direct measure of concentration and the method is semi-quantitative. In principle it could be possible to measure the nonlinear susceptibility of the compound and derive a quantitative value by comparison with a calibration standard (see Supplementary Note 6), or measure the yield at the end of the synthesis to scale the running concentration. If multiple structures with different form factor or nonlinear susceptibility $\chi^{(2)}$ generate an AR-SHS signal the extracted scaling factor will be a convolution of different contributions. As mentioned in the previous paragraph, size distributions can be estimated by analyzing the scattering pattern at higher angles. If structures of different symmetry are present, the relative scattering efficiencies for the different structures matter. In this case the depolarization will appear as a convolution as well, which may help estimating relative concentrations. The evolution of the number of particles over the course of the measurement is shown in the second panel of Fig. 3c. Initially the number of particles progressively decreases until at ~8 min a minimum is reached. These results are in excellent agreement with observations made by static light scattering on the same system by Cravillon et al., who found that before 8 min, smaller particles grow by coalescence after which larger particles grow by monomer addition[37]. This is reflected in the evolution of particle size as well, increasing more rapidly before the change in growth mechanism around ~8 min. The slight increase in number of particles after 8 min implies that nucleation continues during the initial growth phase, also in line with previous crystal growth studies of ZIF-8[37,47]. Since AR-SHS probes the formation of crystalline ZIF-8 in terms of concentration and size the relative crystallinity or extent of crystallization can be plotted against time. A measure of the total mass of ZIF-8 is found by multiplying the number of particles by the particle radius to the third power. Models such as the classical Avrami model can be applied to extract kinetic information[48]. A fit to this model is shown in Supplementary Figure 8 and allowed determination of the Avrami exponent and an overall rate constant. An Avrami exponent of $2.70 \pm 0.09$ results, which is lower than the value of 4 expected for homogeneous nucleation in 3 dimensions. This lowering can reflect the lowering of the dimensionality of space in which crystallization occurs or can be caused by an inhomogeneous distribution of nuclei[49], and has been observed for the formate modulated crystallization of ZIF-8 in solvothermal conditions as well[50]. The data before 8 min show a clear deviation from the fitted curve: the total mass remains rather constant before 8 min, after which it increases exponentially. At the same time the particle radius increases rapidly before 8 min. This is in line with a change in crystallization mechanism

from particle coalescence to growth by monomer addition as discussed above.

Finally note that further adaptations to the AR-SHS technique can be expected to improve the sensitivity of the technique and the information that can be derived. Firstly, the sensitivity of the presented dAR-SHS measurement scheme can still be improved by combining an amplified kHz femtosecond system with a gated intensified charge coupled device camera, which results in large improvements of the signal to noise ratio[36]. Another approach would be to combine dAR-SHS with standard SHS, for instance by sending backscattered SHS collected by the focusing lens to a spectral setup by use of a dichroic mirror. In the earliest stages of nucleation and growth, where small particles show little to no angle dependence and signal levels are low, this approach can boost sensitivity. Secondly, since AR-SHS is a second-order nonlinear process it only detects noncentrosymmetric structures. This can hamper a complete understanding of crystallization processes since potential centrosymmetric phases or intermediates would go unnoticed. However, odd-order techniques such as classical light scattering (first order) or the recently developed third harmonic light scattering (third order) can detect structures of all symmetries[51]. By combining AR-SHS with an odd-order optical technique such as classical light scattering, centric structures would show up in the odd order but not in the AR-SHS signal. Simultaneous measurement of the angle dependence for both techniques could even allow detection of different domain sizes for e.g., amorphous and (noncentrosymmetric) crystalline regions. One way of achieving this would be to select different wavelengths for detection in the back focal plane, analogous to the approach for polarization selection presented in this work. Combination of second and third harmonic light scattering is particularly interesting in this sense since, unlike for linear light scattering, the polarization dependence of the third harmonic relates to the symmetry of the scattering particles[52,53].

## Discussion

The measurement geometry presented here uniquely allows measurement of the SHS signal over a wide range of angles and for different polarization combinations simultaneously. Unlike the current method of measuring AR-SHS patterns, i.e., moving a goniometric arm in discrete steps around the cuvette, dAR-SHS does not require any mechanically moving parts and is therefore inherently fast. This unlocks a wealth of information unattainable by earlier dynamic single-angle experimental configurations. For the MOF ZIF-8, we were able to detect the formation of pure phase ZIF-8 ($T_d$ symmetry), while simultaneously measuring size, shape, and concentration with a time resolution of 3 s. These results highlight the potential of dAR-SHS for studying crystal formation in situ. The accessibility of structural information offers clear advantages over standard optical scattering techniques such as static and dynamic light scattering (SLS/DLS) on one hand. X-ray (or neutron) scattering studies on the other hand can give a comprehensive insight into crystal formation processes, but these techniques generally require bright large-scale synchrotron sources to provide the necessary time resolution, making them inaccessible and expensive. Moreover, extracting structural information through wide-angle X-ray scattering measurements requires relatively large domains having sufficient crystalline order[1]. Structural information obtained through AR-SHS is relatively limited compared to X-ray methods, but the accessibility, experimental cost, and sensitivity of this benchtop technique are clearly advantageous. Additionally, SHS techniques can provide symmetry information from the earliest stages onwards, for individual molecules, aggregates, and small crystallites, i.e.,

before the size/order threshold for WAXS detection is reached. dAR-SHS can thus provide important insights into the nucleation and formation of a wide range of acentric crystalline solids. For pharmaceutical compounds for instance, which are for the largest part acentric (chiral), crystallization is one of the most important operations for the separation, purification and formulation of active molecular compounds[54,55]. In the biological realm, a variety of structures such as collagen fibers, myosin, and micro-tubules are noncentrosymmetric both at molecular and supra-molecular levels[56]. These structures show strong second harmonic signals, and exhibit dynamics (assembly–disassembly) that still need to be elucidated[57,58]. In addition, almost all amino acids lack inversion symmetry and the proteins as well. For centrosymmetric crystals, an intriguing prospect is the possibility to monitor crystal formation exclusively at the surface, where bulk centrosymmetry is broken by default. Finally, besides studying crystal growth, dAR-SHS can be of importance for the study of e.g., adsorption kinetics, solubilization, transport phe-nomena through membranes of different size in cells (with outer membrane vs. organelle membranes resulting in different angular dependence), among myriad other applications. Thus, we believe that our Fourier-imaging approach to AR-SHS will expand its addressable application space by substantially improving the temporal resolution while simultaneously lowering the cost and complexity of the instrumentation.

## Methods

**Materials**. Polystyrene beads coated with sulfate groups were purchased from Interfacial Dynamics Corporation (White aldehyde/sulfate terminated, sizes 170, 320, and 490 nm) and brought into aqueous solution at a concentration of $10^6$–$10^7$ particles per ml. The pH was then brought to 4.1 by addition of a 0.1 M HCl solution (Fisher). Malachite green chloride (Sigma Aldrich) was added from a concentrated aqueous stock solution (Milli Q, Millipore, 18 MΩ) to reach a final concentration of 5 μM, which is in excess of the concentration needed to reach maximal surface adsorption density[59]. ZIF-8 was synthesized in methanol (Fisher, analytical grade) according to a published procedure in presence of formate[37]. The molar ratios of the end solution were 1:4:4:1000 for Zn: 2-methylimidazole: sodium formate: methanol. Two stock solutions were mixed to start the synthesis, one containing the zinc source ($Zn(NO_3)_2 \cdot 6H_2O$, Sigma Aldrich) and another con-taining sodium formate (Sigma Aldrich) and 2-methylimidazole (Sigma Aldrich). The measurement cuvette was coated with a PEG-silane layer by immersion in a solution of 1% (v/v) 2-[methoxy (polyethyleneoxy) propyl]—trimethoxysilane (6–9 units, AB111226, abcr) in 90/10 water/methanol for 2 h. After thorough cleaning the cuvette was baked at 110 °C for 15 min. This coating has been applied to avoid heterogeneous crystallization at the cuvette walls, which was confirmed by eye and by the lack of AR-SHS signal of the empty cuvette after crystallization. In our experiments, mixing was achieved outside of the setup by adding the solution of zinc salt to the linker after which the cuvette was closed and shaken vigorously for about 5 s. Thereafter the cuvette was placed in the setup. The moment of addition was taken as $t = 0$. The dead-time due to mixing and transfer of the cuvette was about 40 s. Since for this synthesis an induction time of several minutes was observed here and in earlier studies[37], this procedure did not result in loss of information. For faster procedures, in-cuvette mixing or automated stopped-flow devices can be used.

**Dynamic light scattering measurements**. Time-resolved monitoring of MOF growth using dynamic light scattering was performed with a Zetasizer Nano ZS (Malvern instruments, Malvern, United Kingdom; 4 mW maximum power, 633 nm wavelength laser, measurement angle of 175°). Correlograms were averaged for 1 min per measurement. Data treatment was performed with Malvern Zetasizer software to extract intensity and number distributions. Viscosity values used to calculate particle radii were measured with an AND SV-10 vibroviscometer for the same sample composition and reaction time.

**Optical instrumentation and calibration**. A high repetition (80 MHz), broadband (680–1300 nm) femtosecond pulsed laser (Spectra-Physics InSight DS+), produc-ing a p-polarized laser beam, was used in all experiments. Intensity variation was accomplished by an achromatic half-wave plate followed by a polarizer in the S-direction located immediately after the laser system. Consequently, the polarization used for all experiments for the input beam was S-polarized. A wavelength of 1030 nm was used.

The setup is shown schematically in Fig. 1. The laser beam is focused in a 2 mm path length amorphous quartz cuvette (Hellma, 110–2–40) with a 11 mm aspheric lens (A220TM, Thorlabs). The input intensity was kept at 400 mW for

all experiments. We aligned the input lens so that the focus was in the middle of the cuvette. For the laser beam diameter used here (~2 mm) this results in a spot size of 7.2 μm and a Rayleigh length of about 159 μm. Approaching the beam as the frustum of a cone at both sides of the focus gives a focal volume ~76.2 pL. Given the relatively small Rayleigh length and the fact that AR-SHS depends on the input intensity squared, restricting the generated light to the focal volume, any contribution from the cuvette walls is avoided. The incident laser light propagation direction is aligned at an angle with the optical axis of the collection system, both in a plane horizontal to the optical table. The angle of incidence is larger than 30° that has two advantages. Firstly, since AR-SHS patterns are symmetric around 0° alignment in the forward direction would result in measuring the same portion of the scattering pattern twice around 0°. Secondly, by measuring under an angle the laser does not directly enter the collection system minimizing the risk of artefacts. Note that for an open beam without sample, no light was detected by the camera. The detection is realized by simultaneously imaging a large portion of the scattering patterns on a CCD camera in a Fourier-imaging scheme[38–40], resulting in fast acquisition times. The scattered light is collected by a fast collection lens (L1, Schneider O.95/17 mm). At the Fourier plane (back focal plane) of this lens an image of the scattering pattern is formed. A polarizing sheet can be inserted here to select for a certain polarization (S or P as indicated in Fig. 1). A short pass filter (Schott, KG5) is also inserted behind this lens to remove any unwanted laser light. Subsequently the Fourier plane is imaged by a 4 f optical system (L2, $f = 50$ mm, f/1.4, Canon lens FD; L3, $f = 8$ mm, f/1.4, Edmund Optics) to project the scattering pattern on the chip of our EM-CCD camera (Andor, Ixon 897). In front of the last lens (L3), a bandpass filter (Semrock, FF01-517/20-25) is inserted to select for the second harmonic light. Since bandpass filters are angle sensitive, we specifically selected a bandpass filter that passes the second harmonic light up to an angle of incidence of 20°, larger than the cone half angle created by the second lens (L2, $f = 1.4$). The alignment procedure is specified further in the Supplementary Note 1.

**Spectral SHS measurements**. The same laser system and focusing lens as for the AR-SHS measurements are used for spectral SHS measurements during the crys-tallization of ZIF-8. The setup and data-analysis is described in full detail else-where[51], only here a Wollaston prism was added in front of the slit of the spectrometer to separate the second harmonic light in its PSS and SSS contribu-tions which could be detected simultaneously by appropriate binning of the EM-CCD camera. Over the course of the measurement the intensity was varied either to improve the S/N or to avoid overload of the EM-CCD detector. The initial intensity was about 700 mW, thus higher than for our AR-SHS experiments. The data were afterwards corrected for the change in fundamental intensity by assuming a quadratic dependence of the second harmonic signal on the input intensity, resulting in smooth curves, as can be seen in Fig. SI-8. This at the same time implies that no significant higher-order effects such as self-focusing/defocusing or thermal lensing are of importance here, since this would lead to an intensity dependent deviation for the quadratic dependency of the second harmonic.

**Data availability**. The data that support the findings of this study are available from the corresponding author S.V. upon reasonable request.

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

## Acknowledgements

S.V. thanks the Fund Scientific Flanders (FWO-Vlaanderen) for a postdoctoral fellowship. T.V. acknowledges funding from the KU Leuven (Grant C16/16/003). Z.J.S. gratefully acknowledges support from the Natural Science Foundation of China's 1000 Young Talents Global Recruitment Plan.

## Author contributions

M.V., S.V. and T.V. designed the project. Z.J.S. and S.W. assisted with designing and calibrating the optical setup. S.V. performed the dAR-SHS experiments. O.D. and C.B.

measured and analyzed the dynamic light scattering data. S.V. prepared the manuscript. All authors contributed to finalizing the manuscript.

## Additional information

**Competing interests:** The authors declare no competing interests.

