## [Peer Review File · Nature Communications]

Reviewers' comments:

Reviewer #1 (Remarks to the Author):

The major claim of the paper is the development of an in situ measuring technique that enables the simultaneous measurement of particle number density, particle size, particle shape and structural information. In this way particle formation processes, such as the crystallization of a metal organic framework can be followed using unprecedented analytical information. A technique that measures all does not exist yet and therefore is very novel and is of interest well beyond the metal organic framework research field. Also for instance crystallization of chiral compounds could be monitored in this way. Such a technique allows therefore access to information that might lead to an enhanced insight in the nucleation and growth processes of various crystallizing compounds. While the technique seems very interesting, shows high potential and is worth publishing in this journal, I think the analysis of the metal organic framework crystallization experiment is not fully convincing. A more in depth and structured discussion of the implications of the obtained data from the analysis could improve this. The analysis, summarized in figure 4c, shows radius, number of particles and depolarization containing average particle structure information. It appears that the significance of the data in the first few minutes is rather low with large error bars, attributed to low signal to noise ratio. However, why is there a low signal to noise ratio? Is this because of the small particle density/small amount of particle mass present? What is the detection limit that arises from these experiments? Usually scattering enables the determination of particle size distributions. However, here only an average size is given. Is distribution information hidden in the data to be analyzed or is it not there? The number of particles is reported in an arbitrary unit and is stated to need a calibration (how?). In the analysis shape is assumed spherical while shape is mentioned as one of the 4 properties monitored in the abstract.

It is not reported what the implications of the structural information measurements are. Does this mean that there are no particles with another structure present or precede the formation of ZIF-8? In the discussion section one would expect how this measurement of structure can be used to obtain information on the early stages of a crystallization process.

The last paragraph of section 2 could better be split into 4, all dealing with one of the properties measured.

It would be good to structure the discussion into paragraphs with an overall look at the data generated by the new technique as well as a paragraph looking into the future.

Other comments:

Page 1, line 29: what insight is provided? "Insight" indicates gain of understanding which would come from the interpretation of the data shown in figure 4c.

Page 1, line 35: I know the term Crystal Engineering as the design of crystal structures. Is that appropriate here?

Page 2, line 42: growth process should be crystallization process.

Why is particle radius and not particle size plotted in figure 4c?

The term concentration can be better defined in the paper.

Some wording could be adjusted:

Line 66: measurements measuring

Line 120: the use of you

Line 205: sufficiently - substantially

We would also be grateful if you could comment on the appropriateness and validity of any statistical analysis, as well the ability of a researcher to reproduce the work, given the level of detail provided.

Line 239: I don't understand the sentence.

Line 242: Phase pure – pure phase

Line 243-244: twice is found

Line 246: picks up – is significant

Reviewer #2 (Remarks to the Author):

In the present manuscript a method is described to measure second harmonic scattering from growing crystallites in solution. This is an interesting way to monitor crystallization and I would probably recommend publication after several changes have been made.

At present the layout of the manuscript does not support the best possible presentation of the experiment, and the description is at points too vague.

Details:

The manuscript could be written more clearly and with a higher emphasis on the novelty of the work: the time laps measurement of crystal growth in solution. At present there is a lot of text spend on the calibration (PS + MG data), which is not so interesting from a science perspective and not so much text is spend on the actual materials science: the description of the data and the used models are in the SI, the concept of depolarization is not explained and how it might relate to the actual structure of the crystal. I would therefore suggest to move the calibration part mostly to the SI and putting a more detailed description of the actual ZIF data and how it can be treated. The space group and the consequences for the non-zero tensor elements needs to be explained more clearly (is there only one non-zero element?); how the depolarization connects to these tensor elements and what can be said about the crystals from it needs to be discussed more clearly. What are the input parameters for the model, and what would be known from literature.

In addition to the presentation there are the following questions:

The focal volumes of the employed lenses are not given, as is the diameter of the focus and Rayleigh length, so that a reader may justify that the made comments about the dimensions of the experiments are chosen correctly.

Is there thermal lensing? Is the SH process really a second-order process or are there higher order effects? The SSS pattern suggests that this could be the case (even though the authors attribute it to artifacts of the imaging system).

Why is the image in fig 2 not symmetric in the Y direction (vertical)? The top part is darker than the bottom part.

What is the effect of off-axis beams propagating through the input and output windows of the cuvette? Does it affect the scattering pattern?

Can we learn anything on the crystallization kinetics from the derived parameters?

What are the assumptions on the size distribution of the crystallites and why is the number density in a.u.? Should it not be possible to derive actual concentrations of particles?

How would the method work if centrosymmetric crystals are formed?

Why is the DLS size data and SHS size data different? They both have the same size dependence, so the R scaling would produce identical errors in both measurements.

Are there other polarization combinations useful, for example to independently learn something from the surface of the crystals?

In terms of nomenclature all previous SHG works (Heinz, Yang, Gonella, Roke) use the

terminology Angle Resolved instead of angle dependent. To limit confusion in the field I suggest to adopt the term angle resolved (AR) instead of AD.

Several papers by Gonella report on difficulties to describe the PS-MG scattering response with the RGD model. Are these difficulties also apparent here? Again, the description here does give fit functions (fig 2) but no parameters are given. How can a reader judge the procedure if such details are missing?

Response to reviewers

First and foremost the authors thank the reviewers for their insightful comments. The body of the text concerning the data-analysis has been extended, improving the overall quality and clarity of the article. Since it was suggested to rewrite part of the results section of the article, substantial changes have been made both to the manuscript and the supporting information. We have also changed the title to:

“In situ monitoring of symmetry, size, shape and concentration during crystal growth of ZIF-8 with dynamic angle-resolved second harmonic scattering”

so that scientists interested in the crystal growth of ZIF-8 can find their way to our work more easily. That being said, we have first and foremost tried to focus the discussion on the potential of the AR-SHS technique, rather than the material under study, which is why several graphs and details were moved to the supporting information.

The corrected manuscripts were included, for which color coding indicates which adaptations have been made to the previous version, as well as a final new version for which all the changes have been incorporated to improve the readability for the reviewers. In what follows we will answer to the specific questions raised by the reviewers.

Reviewers' comments:

Reviewer #1 (Remarks to the Author):

While the technique seems very interesting, shows high potential and is worth publishing in this journal, I think the analysis of the metal organic framework crystallization experiment is not fully convincing. A more in depth and structured discussion of the implications of the obtained data from the analysis could improve this. The analysis, summarized in figure 4c, shows radius, number of particles and depolarization containing average particle structure information. It appears that the significance of the data in the first few minutes is rather low with large error bars, attributed to low signal to noise ratio. However, why is there a low signal to noise ratio? Is this because of the small particle density/small amount of particle mass present? What is the detection limit that arises from these experiments? Usually scattering enables the determination of particle size distributions. However, here only an average size is given. Is distribution information hidden in the data to be analyzed or is it not there? The number of particles is reported in an arbitrary unit and is stated to need a calibration (how?). In the analysis shape is assumed spherical while shape is mentioned as one of the 4 properties monitored in the abstract. It is not reported what the implications of the structural information measurements are. Does this mean that there are no particles with another structure present or precede the formation of ZIF-8? In the discussion section one would expect how this measurement of structure can be used to obtain information on the early stages of a crystallization process. The last paragraph of section 2 could better be split into 4, all dealing with one of the properties measured.

It would be good to structure the discussion into paragraphs with an overall look at the data generated by the new technique as well as a paragraph looking into the future.

Answer: The entire results section was rewritten as suggested. All questions concerning the data-analysis have been addressed in the added paragraphs dealing with analysis towards i) structural information, ii) particle size, iii) shape and dispersity, iv) number of particles and v) future prospects. We also adapted the definition of depolarization to $I(\text{PSS})/I(\text{SSS})$, instead of $I(\text{SSS})/I(\text{PSS})$, to be in line with literature.

It appears that the significance of the data in the first few minutes is rather low with large error bars, attributed to low signal to noise ratio. However, why is there a low signal to noise ratio? Is this because of the small particle density/small amount of particle mass present? What is the detection limit that arises from these experiments?

The question of sensitivity in the beginning of the measurements is a valid one, since DLS is able to detect the first particles sooner (after ~2 minutes). In the beginning of the measurement the fitting towards the particle radius was not significantly improved by averaging and smoothing our data (average over 10 measurements, smoothed by 5-point adjacent averaging). We therefore addressed the question of sensitivity by performing a traditional second harmonic scattering (SHS) experiment (see paragraph 'Particle size' and section SI-6). Since in a traditional setup the signal is collected over a wide range of angles (high NA collection optics) instead of being spread out to different angles such as for AR-SHS, the sensitivity is markedly higher. Indeed, we detect a measurable background signal from the starting solution before the onset of crystallization. The limit of detection is in this case determined by the relative signal of the forming nuclei/ particles compared to the background. In our experiment after about 2 minutes an increase in signal is witnessed, around the same time as for the DLS technique. The depolarization almost immediately tends towards the expected value for T_d as for the AR-SHS results, implying that the first detected particles by DLS are indeed of ZIF-8 symmetry. In the paragraph concerning future prospects we suggest different ways to further improve the sensitivity, for instance by combining traditional SHS with AR-SHS.

Does this mean that there are no particles with another structure present or precede the formation of ZIF-8?

In the last paragraph of the results section we address this question. AR-SHS is only sensitive to noncentrosymmetric structures, so we cannot exclude formation of centric structures or intermediates. Combination of techniques based on odd (e.g. 'linear' light scattering, detects structures of all symmetries) and even order processes (e.g. SHS, detects acentric structures) is recommended for a full understanding of crystallization, we outlined this in the new version of the paper as well. By comparison of the DLS data with the aforementioned traditional SHS experiment, we find that the first increase in SHS intensity occurs at the same moment the first particles are detected by DLS. Moreover, as

mentioned above, the depolarization immediately tends towards the value expected for ZIF-8 (T_d). This implies that the first detected particles have ZIF-8 symmetry.

Other comments:

Page 1, line 29: what insight is provided? "Insight" indicates gain of understanding which would come from the interpretation of the data shown in figure 4c.

Firstly, the observation that particles adopt ZIF-8 symmetry from the earliest stages of crystallization in this crystallization has not been observed previously. Secondly, we found strong indications supporting the mechanism of coalescence followed by growth by monomer addition proposed by Cravillon and co-workers (reference 43) based on static light scattering measurements, as discussed in the paragraph about 'number of particles'.

We added following sentence to the abstract:

"The angle dependence of the AD-SHS signal provided insight in the growth mechanism by probing the evolution of size, shape and concentration, while polarized measurements yield structural information in terms of point group symmetry."

Page 1, line 35: I know the term Crystal Engineering as the design of crystal structures. Is that appropriate here?

Rightly so, we changed the two first sentences to:

"Gaining control over crystallization processes is a longstanding ambition of crystal growers. Hitherto process optimization often relies exclusively on characterization of the final product, assuming that this gives an unambiguous connection to solution processes such as nucleation and growth.¹"

Page 2, line 42: growth process should be crystallization process.

Adapted as suggested.

Why is particle radius and not particle size plotted in figure 4c?

We chose to plot particle radius in line with the model and earlier work of de Beer et al. (reference 34).

The term concentration can be better defined in the paper.

A paragraph was added discussing the implications of the 'number of particles' N extracted from the model of de Beer et al. (reference 34).

Some wording could be adjusted:

Line 66: measurements measuring

Line 120: the use of you

Line 205: sufficiently – substantially

Adapted as suggested.

We would also be grateful if you could comment on the appropriateness and validity of any statistical analysis, as well the ability of a researcher to reproduce the work, given the level of detail provided.

The amount of detail about the data analysis that was added in response to the referees useful comments have definitely provided sufficient detail to reproduce the work, and of course we would highly appreciate if other researchers implement our technique as well. In the SI details about alignment and data treatment are therefore also provided. All data fitting was performed using least squares, assuming zero-mean Gaussian-distributed errors as would be expected when camera and other system noise are the dominant error sources. We have also included fit parameters where necessary in the SI. The gathered data are also available to other researchers in line with the guidelines of Nature Communications. Regarding the reproducibility of the measurement, there is an intrinsically variable nature to crystallization processes as rare event phenomena. Hence a series of repeat experiments is more likely to provide insight into variability in the crystallization process, and in our specific case about the variability of the reagent mixing procedure, than to provide an experimental error analysis. Two repetitions of the experiments can be found in section SI-9, they resulted in similar curves as those shown in the main paper.

Line 239: I don't understand the sentence.

Line 242: Phase pure – pure phase

Line 243-244: twice is found

Line 246: picks up – is significant

Adapted as suggested.

Reviewer #2 (Remarks to the Author):

In the present manuscript a method is described to measure second harmonic scattering from growing crystallites in solution. This is an interesting way to monitor crystallization and I would probably recommend publication after several changes have been made. At present the layout of the manuscript does not supports the best possible presentation of the experiment, and the description is at points too vague.

Details:

The manuscript could be written more clearly and with a higher emphasis on the novelty of

the work: the time laps measurement of crystal growth in solution. At present there is a lot of text spend on the calibration (PS + MG data), which is not so interesting from a science perspective and not so much text is spend on the actual materials science: the description of the data and the used models are in the SI, the concept of depolarization is not explained and how it might relate to the actual structure of the crystal. I would therefore suggest to move the calibration part mostly to the SI and putting a more detailed description of the actual ZIF data and how it can be treated. The space group and the consequences for the non-zero tensor elements needs to be explained more clearly (is there only one non-zero element?); how the depolarization connects to these tensor elements and what can be said about the crystals from it needs to be discussed more clearly. What are the input parameters for the model, and what would be known from literature.

As suggested I moved most of the calibration part to the SI, and discussed the data analysis and the concept of depolarization in more detail in the different paragraphs (see above) added to the results section. The full derivation of the model for the tensor elements of the symmetry group probed here is given in the SI.

In addition to the presentation there are the following questions:

The focal volumes of the employed lenses are not given, as is the diameter of the focus and Rayleigh length, so that a reader may justify that the made comments about the dimensions of the experiments are chosen correctly.

We have added following sentences in the methods section:

'For the laser beam diameter used here (~2mm) this results in a spot size of 7.2 micrometer and a Rayleigh length of about 159 micrometer. Approaching the beam as the frustum of a cone at both sides of the focus gives a focal volume ~76.2 pL.'

Is there thermal lensing? Is the SH process really a second-order process or are there higher order effects? The SSS pattern suggests that this could be the case (even though the authors attribute it to artifacts of the imaging system).

As mentioned above we have performed standard second harmonic scattering measurements. These measurements were performed at higher input power than for the AR-SHS and the intensity during the measurement was adapted, with results agreeing exactly with the expected quadratic dependence of the second harmonic, ruling out higher order or thermal effects. In the methods section we have added following sentences:

'Over the course of the measurement the intensity was varied either to improve the S/N or to avoid overload of the EM-CCD detector. The initial intensity was about 700 mW, thus higher than for our

AR-SHS experiments. The data were afterwards corrected for the change in fundamental intensity by assuming a quadratic dependence of the second harmonic signal on the input intensity, resulting in smooth curves, as can be seen in Fig. SI-8. This at the same time implies that no significant higher-order effects such as self-focusing/defocusing or thermal lensing are of importance here, since this would lead to an intensity dependent deviation for the quadratic dependency of the second harmonic.'

The figure in the SI (Fig. SI-8) moreover shows the data before and after the correction was applied.

Why is the image in fig 2 not symmetric in the Y direction (vertical)? The top part is darker than the bottom part.

In Fig. 2b a polarizer was added in the back focal plane of the collection lens in order to simultaneously measure different polarization combinations. In the figure caption following sentence has been added:

"Image of the AR-SHS scattering pattern for 320 nm polystyrene beads coated with malachite green with insertion of a polarizing sheet (S-out) in the lower half of the back focal plane. By symmetry considerations the SSS signal vanishes for PS+MG beads,⁴³ which is why the signal vanishes in the bottom half of the image."

What is the effect of off-axis beams propagating through the input and output windows of the cuvette? Does it affect the scattering pattern?

If the windows are plane-parallel, then the windows can be ignored, they only translate the beam without changing its angle, and in our system angle is all we care about. Of course the cuvette walls cannot be perfectly parallel, but any slight non-parallelism in the windows would lead to extremely small changes in the mapping of camera pixel to scattering angle. Basically if the windows had a wedge angle α , the resulting scattering pattern will be linearly shifted in the Fourier plane by a factor of $\alpha \times (1/n_c)$ where n_c is the refractive index of the cuvette. Thus, 1 degree of wedge (which would be quite large, most cuvettes are made with very tight tolerances, see Clinical Chemistry vol. 19 no. 9 1053-1057) would lead to only a ~ 0.66 degree shift in the pattern.

Can we learn anything on the crystallization kinetics from the derived parameters?

We have added a kinetic analysis based on the Avrami model, which is elaborated in the paragraph dealing with 'Number of particles' and in section SI-7.

What are the assumptions on the size distribution of the crystallites and why is the number density in a.u.? Should it not be possible to derive actual concentrations of particles?

A paragraph discussing determination of size distributions was added (paragraph 'shape and dispersity'). The matter of absolute calibration is discussed in the paragraph dealing with 'Number of particles' and in more detail in the SI (section SI.8).

How would the method work if centrosymmetric crystals are formed?

As discussed above in response to the question of reviewer 1, this issue is addressed in the last paragraph of the Results section. For centrosymmetric crystals it would in principle be possible to measure a surface contribution, although sensitivity is expected to be an issue here, especially for smaller crystallites. However, improvements in sensitivity, also discussed in the same paragraph, might produce a measurable response. We also touched upon this possibility in the discussion section.

Why is the DLS size data and SHS size data different? They both have the same size dependence, so the R scaling would produce identical errors in both measurements.

SHS and DLS data have been measured in repeated experiments, so not simultaneously. Some variation can therefore be expected between different crystallization experiments. On the other hand, the change in mechanism from coalescence to monomer growth witnessed by static light scattering experiments by Cravillon et al. (ref. 37) manifests itself in the AR-SHS data as a more rapid change in particle radius before ~8 minutes. This is not apparent from the DLS data at that point. Of course, DLS probes the diffusion motion which means that small aggregates like doublets, triplets, etc., would be seen as "particles" of 2x or 3x the volume. In contrast, SLS would see the pattern of the small particles overlaid with an interference pattern related to the inter-particle distance in the aggregates. In the limit of a large number of such aggregates where there are a large number of inter-particle distances and orientations, these "overlaid" interference patterns would tend to wash out, and one would be left with the pattern of the small scatterers only. This might be the origin of the apparent difference between both observations.

Are there other polarization combinations useful, for example to independently learn something from the surface of the crystals?

I added following sentence to the second paragraph of the results section:

'Note that depending on the system under study, other polarization combinations can provide additional information, for instance on the chirality of the structure.⁴¹

To independently learn something from the surface of the crystals is possibly not realistic since the bulk signal will dominate the surface contribution.

In terms of nomenclature all previous SHG works (Heinz, Yang, Gonella, Roke) use the terminology Angle Resolved instead of angle dependent. To limit confusion in the field I suggest to adopt the term angle resolved (AR) instead of AD.

We have adapted the terminology as suggested.

papers by Gonella report on difficulties to describe the PS-MG scattering response with the RGD model. Are these difficulties also apparent here? Again, the description here does give

fit functions (fig 2) but no parameters are given. How can a reader judge the procedure if such details are missing?

We added the full in section SI-3. The model reported by Yang et al. shows good results for particles smaller than the wavelength. We see good correspondence as well, therefore we did not seek to employ more intricate models.

REVIEWERS' COMMENTS:

Reviewer #1 (Remarks to the Author):

The authors have adequately acted on the reviewer comments. I have no further comments on this highly interesting paper.

Reviewer #2 (Remarks to the Author):

The manuscript has much improved and I recommend publication. Nice work!